# LDLR, LRP1, and Megalin redundantly participate in the uptake of *Clostridium novyi* alpha-toxin

Yao Zhou[1,2,3,4,5], Danyang Li [1,2,3,4,5], Diyin Li[1,2,3,4], Aizhong Chen[2,3,4], Liuqing He [2,3,4], Jianhua Luo [2,3,4] & Liang Tao [1,2,3,4✉]

*Clostridium novyi* alpha-toxin (Tcnα) is a potent exotoxin that induces severe symptoms including gas gangrene, myositis, necrotic hepatitis, and sepsis. Tcnα binds to sulfated glycosaminoglycans (sGAG) for cell-surface attachment and utilizes low-density lipoprotein receptor (LDLR) for rapid entry. However, it was also shown that Tcnα may use alternative entry receptors other than LDLR. Here, we define that LRP1 and Megalin can also facilitate the cellular entry of Tcnα by employing reconstitutive LDLR family proteins. LDLR, LRP1, and Megalin recognize Tcnα via their ligand-binding domains (also known as LDL receptor type A repeats). Notably, LDLR and LRP1 have contrasting expression levels in many different cells, thus the dominant entry receptor for Tcnα could be cell-type dependent. These findings together increase our knowledge of the Tcnα actions and further help to understand the pathogenesis of *C. novyi* infection-associated diseases.

[1] College of Life Sciences, Zhejiang University, 310058 Hangzhou, Zhejiang, China. [2] Key Laboratory of Structural Biology of Zhejiang Province, School of Life Sciences, Westlake University, 310024 Hangzhou, Zhejiang, China. [3] Center for Infectious Disease Research, Westlake Laboratory of Life Sciences and Biomedicine, 310024 Hangzhou, Zhejiang, China. [4] Institute of Basic Medical Sciences, Westlake Institute for Advanced Study, 310024 Hangzhou, Zhejiang, China. [5] These authors contributed equally: Yao Zhou, Danyang Li. ✉email: taoliang@westlake.edu.cn

*C*lostridium novyi is an anaerobic, motile, and spore-forming bacterium that causes severe infectious diseases in humans and animals including gas gangrene, myositis, necrotic hepatitis, and sepsis[1–3]. *C. novyi* alpha-toxin (Tcnα) is a critical factor found in all pathogenic *C. novyi* strains, which are edematizing and lethal[4]. Tcnα belongs to a structurally related protein family called the large clostridium toxin (LCT) family. Members of the LCT family share similar domain arrangements as well as toxin action mechanisms[5]. All known LCTs (except for TpeL) consist of four functional domains, including a glucosyl-transferase domain (GTD), an autocatalytic cysteine protease domain (APD), a delivery and receptor-binding domain (DRBD), and the combined repetitive oligopeptides (CROPs) domain[5–7]. Like other LCTs, Tcnα binds to the cell-surface receptors and enters cells via endocytosis. The low pH of endosomes induces structural changes in the toxin. The GTD is then delivered across endosomal membranes, released into the cytoplasm, and glucosylates small GTPases of the Rho and Ras family, leading to cytoskeleton disruption and eventual cell death[8,9]. Unlike other LCTs, Tcnα and TpeL are the only two LCTs that use UDP-N-acetylglucosamine (TpeL can also utilize UDP-glucose) to modify targeting small GTPases[10–12].

Previously we reported that low-density lipoprotein receptor (LDLR) mediates the cellular entry of Tcnα[13]. It was also suggested that LDLR family members other than LDLR may participate in the entry of Tcnα because LRPAP1 (also known as RAP), a general binder to LDLR family members, further protected the LDLR KO cells from Tcnα[13]. Core members of the LDLR family, including LDLR, VLDLR, LRP1, Megalin (also known as LRP2 or gp330), ApoER2 (LRP8), LRP1B, and MEGF7, are well-known to mediate the endocytosis of a variety of ligands and maintain internal homeostasis[14]. These proteins have a large extracellular domain, a single transmembrane domain, and a relatively short cytoplasmic tail[15]. The extracellular domains of LDLR family proteins consist of several modular structures,

including LDL receptor type A (LA) repeats, LDL receptor type B (LB) repeats (also known as epidermal growth factor precursor homology regions with β-propeller repeats), and an O-linked sugar domain[15]. Each LA module is about 40-60 residues long and displays a disulfide-bond stabilized charged surface[16,17]. The LA domains are commonly known as the ligand-binding regions recognizing various ligands such as ApoB, ApoE, LRPAP1, and Vesicular stomatitis virus (VSV)[18–21]. The bound ligands are commonly believed to be released in the low pH environment upon endocytosis[22,23]. The Asn-Pro-X-Tyr motif (NPxY; with x representing any amino acid) is found in several LDL receptor family members and can facilitate coated-pit-mediated endocytosis[24].

Although our previous study indicated that LDLR family members other than LDLR may serve as redundant endocytic receptor(s) for Tcnα, the receptor selectivity within the family remains unclear. Here, we examined the contribution of major LDLR family members in the cellular entry of Tcnα by ectopically expressing native or reconstituted proteins in the HeLa *LDLR*−/− cells. We reported that LRP1 and Megalin, but not other tested LDLR family members, could functionally mediate the entry of Tcnα. We also found that LDLR, LRP1, and Megalin have varying expression levels in different cell types, thus Tcnα may use different entry receptors to intoxicate various host cells.

## Results

**The LA repeats of LDLR are responsible for the uptake of Tcnα.** The extracellular domain of LDLR consists of an LA domain, an LB domain, and an O-linked sugar region. To interrogate the regions in LDLR involving the uptake of Tcnα, we generated two Ldlr truncates lacking either the LA repeats domain (Ldlr$_{\Delta LA}$) or the LB repeats domain (Ldlr$_{\Delta LB}$), as well as an Ldlr with its NPxY motif deleted (Fig. 1a). The HeLa WT and *LDLR*−/− were exposed to different concentrations of Tcnα for 3 h.

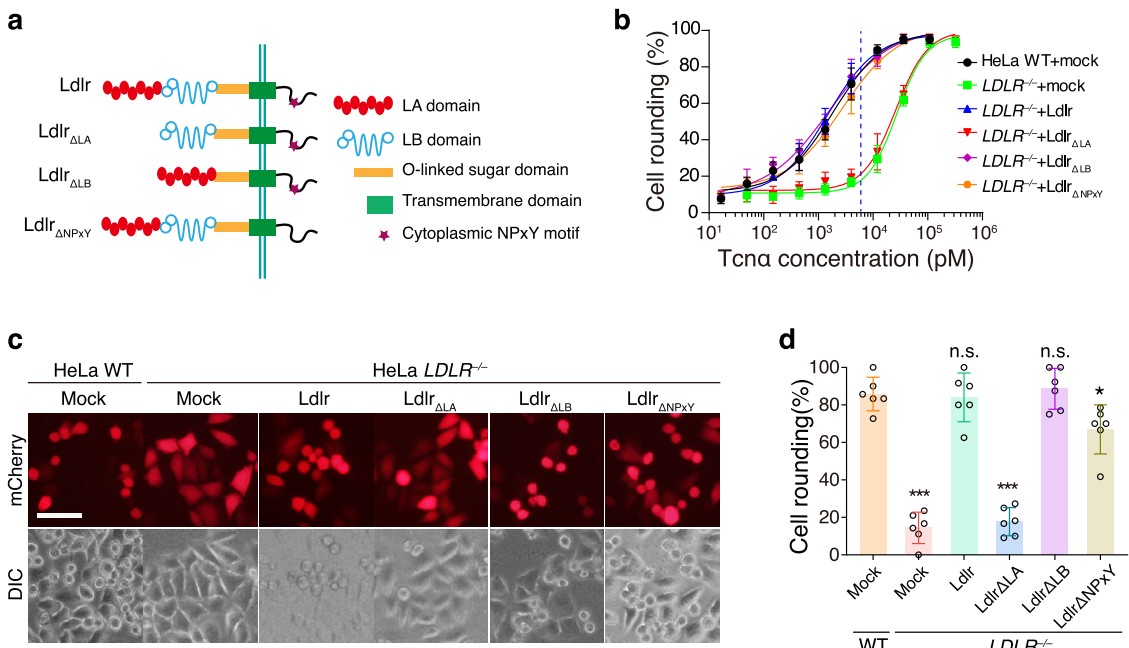

**Fig. 1 The LA domain of LDLR is responsible for the uptake of Tcnα. a** Schematic drawing of Ldlr, Ldlr$_{\Delta LA}$, Ldlr$_{\Delta LB}$, and Ldlr$_{\Delta NPxY}$. **b** The HeLa WT or *LDLR*−/− cells were transfected with mock, Ldlr, Ldlr$_{\Delta LA}$, Ldlr$_{\Delta LB}$, and Ldlr$_{\Delta NPxY}$, followed by incubating with Tcnα for 3 h. The percentages of round-shaped cells are plotted on the chart. The blue dash line indicates 6 nM. Error bars (*n* = 6) indicate mean ± SD. **c** The indicated HeLa cells were incubated with Tcnα (6 nM, 3 h) and the images were captured. Red fluorescence (mCherry) marked transfected cells. The scale bar represents 50 μm. **d** The round-shaped cells among all mCherry-positive cells shown in **c** were quantified and plotted in a bar chart. Error bars (*n* = 6) indicate mean ± SD, *$P < 0.05$, ***$P < 0.001$, n.s. = not significant, two-sided Student's *t*-test.

6 nM Tcnα induced ~80% of the HeLa WT cells to become round in 3 h while the $LDLR^{-/-}$ cells are generally normal (Fig. 1b). This assay condition was adopted for testing the sensitivity of other transfected HeLa cells unless otherwise stated. Ectopic expression of the full-length Ldlr and Ldlr$_{\Delta LB}$, but not Ldlr$_{\Delta LA}$, restored susceptibility of the $LDLR^{-/-}$ cells to Tcnα, suggesting the LA repeats are essential for mediating the entry of Tcnα (Fig. 1b–d and Supplementary Fig. 1a). This data is in line with the previous finding that LRPAP1, which binds to the LA domain of LDLR[21], can competitively protect cells from Tcnα[13]. Besides, an Ldlr mutant with NPxY motif deleted could restore the susceptibility of $LDLR^{-/-}$ cells but less efficiently (Fig. 1b–d). NPxY motif is responsible for the fast recycling of LDLR[24], which promotes the uptake of the toxin but is not necessary.

**Reconstituted LRP1 and Megalin sensitize the HeLa $LDLR^{-/-}$ cells to Tcnα.** All LDLR family core members contain at least one LA-repeats domain. Because LRPAP1 further protects the LDLR KO cells from Tcnα[13], we postulate that the LA repeats from other LDLR family proteins may also recognize Tcnα. HeLa $LDLR^{-/-}$ cells are more resistant to Tcnα compared to the WT cells and ectopic expression of a mouse Ldlr would restore their susceptibility. This cell system could be used for investigating other potential endocytic receptors of Tcnα. LRP1, Megalin, and LRP1B are very large proteins (~600 kDa) that are hard to be expressed. The extracellular domains of both LRP1 and Megalin contain four canonical LA repeats domains, namely cluster I-IV, with clusters II and IV particularly important for ligand-binding[25,26]. Therefore, we fused the cluster II LA domains of LRP1, Megalin, and LRP1B to the C-terminal part (including the EGF-precursor domain, O-linked sugar domain, transmembrane region, and cytoplasmic domain) of Ldlr (Ldlr$_C$) and generated chimeric proteins, including LRP1$_{CII}$-Ldlr$_C$, Megalin$_{CII}$-Ldlr$_C$, and LRP1B$_{CII}$-Ldlr$_C$ (Fig. 2a). To interrogate alternative entry receptor(s) of Tcnα within the LDLR family, Ldlr, Vldlr, ApoER2, Lrp4, Lrp10, Lrp11, LRP1$_{CII}$-Ldlr$_C$, Megalin$_{CII}$-Ldlr$_C$, and LRP1B$_{CII}$-Ldlr$_C$ were exogenously expressed in the HeLa $LDLR^{-/-}$ cells by transient transfection (Supplementary Fig. 1b). The Tcnα sensitivities of these transfected cells were measured by the cytopathic cell rounding assay. Ectopic expression of Ldlr, LRP1$_{CII}$-Ldlr$_C$, and Megalin$_{CII}$-Ldlr$_C$, but not others, sensitized the Hela $LDLR^{-/-}$ cells to Tcnα (Fig. 2b, c). We next switched the C-terminal part of LRP1$_{CII}$-Ldlr$_C$ to LRP1$_C$ (Fig. 2a). As expected, this newly built LRP1$_{CII}$-LRP1$_C$ also effectively mediates the entry of Tcnα (Fig. 2b, c). These results suggest that the LA domains from LDLR, LRP1, and Megalin can selectively recognize Tcnα.

**Surface sGAG is essential for LDLR/LRP1/Megalin-mediated uptake of Tcnα.** Cell-surface sGAG can mediate the attachment of Tcnα and *Clostridioides difficile* toxin A (TcdA) and allow them to be enriched on the cell surface[13,27]. To demonstrate the sGAG-binding potentials of other major LCTs, we performed the heparin-beads pulldown experiment with the purified LCT proteins. While Tcnα strongly binds to the heparin beads, minimal bindings of TcsH and TpeL were observed, and no TcsL or TcdB binding was detected (Fig. 3a).

The previous study reported that direct interaction between LDLR and Tcnα is weak. Using the biolayer interferometry (BLI) assay, we showed that both interactions between Ldlr$_{LA}$ and Tcnα and between LRP1$_{CII}$ and Tcnα are weak (Supplementary Fig. 2a, b). Routine dot-blot assays showed no detectable signals for LRP1$_{CII}$-Tcnα binding (Supplementary Fig. 2c). However, if the dot-blot assays were performed followed by 1-Ethyl-3-[3-dimethylaminopropyl] carbodiimide hydrochloride (EDC) cross-link[28], obvious

signals for LRP1$_{CII}$/LDLR$_{LA}$ binding to membrane immobilized Tcnα were detected (Supplementary Fig. 2d). These results suggest that the interactions between LRP1$_{CII}$/LDLR$_{LA}$ and Tcnα could be either weak or unstable. To investigate whether surface sGAG promote the LDLR/LRP1/Megalin-mediated cellular entry of Tcnα, we employed HeLa $SLC35B2^{-/-}$ cells that lack sulfation in surface proteoglycans and are thus considered sGAG-negative[27,29]. HeLa $SLC35B2^{-/-}$ cells were transiently transfected with Ldlr, LRP1$_{CII}$-Ldlr$_C$, and Megalin$_{CII}$-Ldlr$_C$, cells transfected with an empty vector served as the controls. These transfected cells were pre-incubated with 200 nM Tcnα on ice for 30 min, changed with the fresh medium, and incubated at 37 °C for 3 h. Overexpression of Ldlr, LRP1$_{CII}$-Ldlr$_C$, or Megalin$_{CII}$-Ldlr$_C$ failed to sensitize the $SLC35B2^{-/-}$ cells to Tcnα (Fig. 3b, c). Together, these data demonstrated that cell-surface sGAG is essential for Ldlr-, LRP1-, and Megalin-mediated cellular entry of Tcnα.

**LRP1 versus LDLR in different cells.** Although LRP1 and Megalin can mediate the cellular entry of Tcnα, they were not found in the candidate list of our previous CRISPR screen for Tcnα[13]. Likewise, LRP1 was demonstrated as an entry receptor for TcdA[30] but it did not stand out from the previous genome-wide screen[27]. We noticed that HeLa cells were employed in both genetic screens for TcdA and Tcnα, as well as the following validation experiments. On the other hand, previously Schottelndreier et al. used mouse embryonic fibroblasts (MEFs) for studying the role of LRP1 in TcdA entry[30]. According to a public protein profiling database (http://www.proteinatlas.org)[31,32], LDLR and LRP1 have contrasting mRNA expression profiles in many different cell lines, while Megalin is absent in most cell lines (Fig. 4a and Supplementary Fig. 3). Interestingly, HeLa cells express LDLR at a high mRNA level and LRP1 at a low mRNA level (Fig. 4a), which may partly explain why LRP1 did not stand out in the previous screens using HeLa cells.

**Both LDLR and LRP1 participate in the Tcnα entry in U-87 MG cells.** We next performed immunoblot analysis to validate the protein levels in some commonly used cell lines including MCF-7, HeLa, HepG2, MEFs, BJ, and U-87 MG. HeLa cells express a minimal amount of LRP1, which is consistent with the mRNA data (Fig. 4a, b). Both MEFs and U-87 MG cells express considerable amounts of LDLR and LRP1 (Fig. 4b). MEFs were previously used to study the role of Lrp1 in mediating the entry of TpeL and TcdA but these are mouse cells[30,33]. Therefore, we chose U-87 MG, a human glioma cell line that expresses both LDLR and LRP1, to generate LDLR and LRP1 knockout cells using the CRISPR/Cas9 approach (Fig. 4c). In this cell line, knocking-out LRP1 does not affect the expression level of LDLR and vice versa (Fig. 4c). We observed that Tcnα is equally bound to the U-87 MG WT, $LDLR^{-/-}$, and $LRP1^{-/-}$ cells in the binding assay (Fig. 4d), which is consistent with the view that LDLR and LRP1 are not dominant attachment factors for Tcnα.

We next assessed the colocalization of the endocytosed Tcnα and LDLR/LRP1 using the toxin internalization assay, followed by confocal fluorescence analysis. In the HeLa cells, knocking out LDLR largely reduced the internalization of Tcnα, indicating that LDLR is a dominant entry receptor for Tcnα in these cells (Supplementary Fig. 4). In contrast, a considerable amount of internalized Tcnα was observed in the U-87 MG $LDLR^{-/-}$ cells using the internalization assay (Fig. 4e). Moreover, the internalized Tcnα better colocalized with LDLR in the U-87 MG $LRP1^{-/-}$ cells when compared to the WT cells (Fig. 4e, f).

Finally, we investigated the roles of LDLR- and LRP1-mediated Tcnα entry and intoxication in the U-87 MG cells. We found that both U-87 MG $LDLR^{-/-}$ and $LRP1^{-/-}$ cells were more resistant to

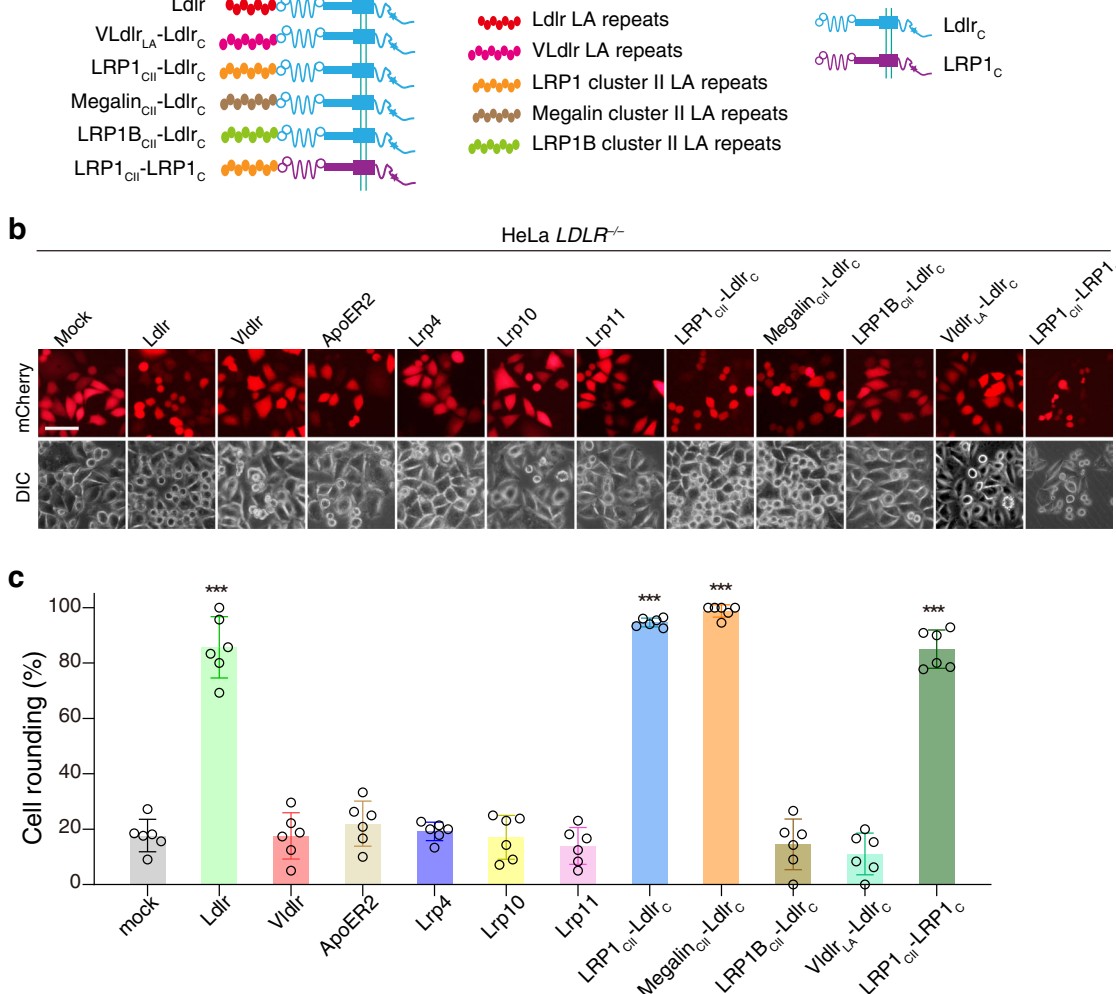

**Fig. 2 LA domains of LRP1 and Megalin recognize Tcnα. a** Schematic drawing of chimeric receptor proteins including VLdlr$_{LA}$-Ldlr$_C$, LRP1$_{CII}$-Ldlr$_C$, Megalin$_{CII}$-Ldlr$_C$, LRP1B$_{CII}$-Ldlr$_C$, and LRP1$_{CII}$-LRP1$_C$. **b** The HeLa *LDLR$^{-/-}$* were transfected with mock, Ldlr, Vldlr, ApoER2, Lrp4, Lrp10, Lrp11, LRP1$_{CII}$-Ldlr$_C$, Megalin$_{CII}$-Ldlr$_C$, LRP1B$_{CII}$-Ldlr$_C$, VLdlr$_{LA}$-Ldlr$_C$, and LRP1$_{CII}$-LRP1$_C$, followed by the incubation with Tcnα (6 nM, 3 h). Red fluorescence (mCherry) marked transfected cells. Representative images are shown. The scale bar represents 50 μm. **c** The round-shaped cells among all mCherry-positive cells shown in **b** were quantified and plotted in a bar chart. Error bars (*n* = 6) indicate mean ± SD, ***P < 0.001 versus mock, two-sided Student's *t*-test.

Tcnα when compared to the WT cells (Fig. 5a). To quantitatively determined the increased resistance, we defined the toxin concentration that results in 50% cell rounding as CR$_{50}$. The CR$_{50}$ for Tcnα in the U-87 MG WT is about 15.8 pM. The *LDLR$^{-/-}$* cells showed ~36-fold increased resistance while the *LRP1$^{-/-}$* cells showed ~18-fold increased resistance, compared to the WT cells (Fig. 5b). While the sensitivity of the *LDLR$^{-/-}$* cells to Tcnα can be restored by the transient transfection of Ldlr, we further showed that ectopic expressing LRP1$_{CII}$-LdlrC restored the sensitivity of the U-87 MG *LRP1$^{-/-}$* cells (Fig. 5c, d). These data together suggest that both LDLR and LRP1 functionally mediate the endocytosis of Tcnα and are redundant receptors for Tcnα in cells such as U-87 MG.

## Discussion

Tcnα is the most important virulence factor responsible for human and animal diseases associated with *C. novyi* infection. Our previous study demonstrated that sGAG and LDLR synergistically mediate the cellular entry of Tcnα[13]. It was also shown that other LDLR family proteins may be redundant entry receptors for Tcnα, but the receptor specificity within the LDLR family remains unclear. However, some LDLR family proteins,

such as LRP1, Megalin, and LRP1B, have very high molecular weights that are hard to be studied directly. Here, we used reconstitutive proteins to investigate the roles of LDLR family members in the cellular uptake of Tcnα. Although the truncated/ chimeric proteins may not completely represent the biological properties of native proteins, they act as powerful tools to study the ligand-binding properties of LDLR family proteins. For example, Ganaie et al. recently used various chimeric LRP1 to investigate the cellular entry of the Rift Valley fever virus[34]. By employing the reconstitutive LDLR family proteins, we successfully defined that LDLR, LRP1, and Megalin serve as redundant entry receptors for Tcnα and their LA domains are responsible for toxin recognition.

LDLR family receptors rapidly and constitutively recycle between cell membranes and endosomes, which provides an ideal route for mediating the endocytosis of target cargoes into cells. Several LDLR family core members commonly share their ability to bind a variety of ligands from endogenous lipoproteins to pathogenic viruses and bacterial toxins, such as LRPAP1, ApoE, TcdA, and vesicular stomatitis virus[21,27,30,35,36]. The LA repeats of the LDLR family core members are closely related modules that are responsible for the binding of most ligands[14]. We also defined

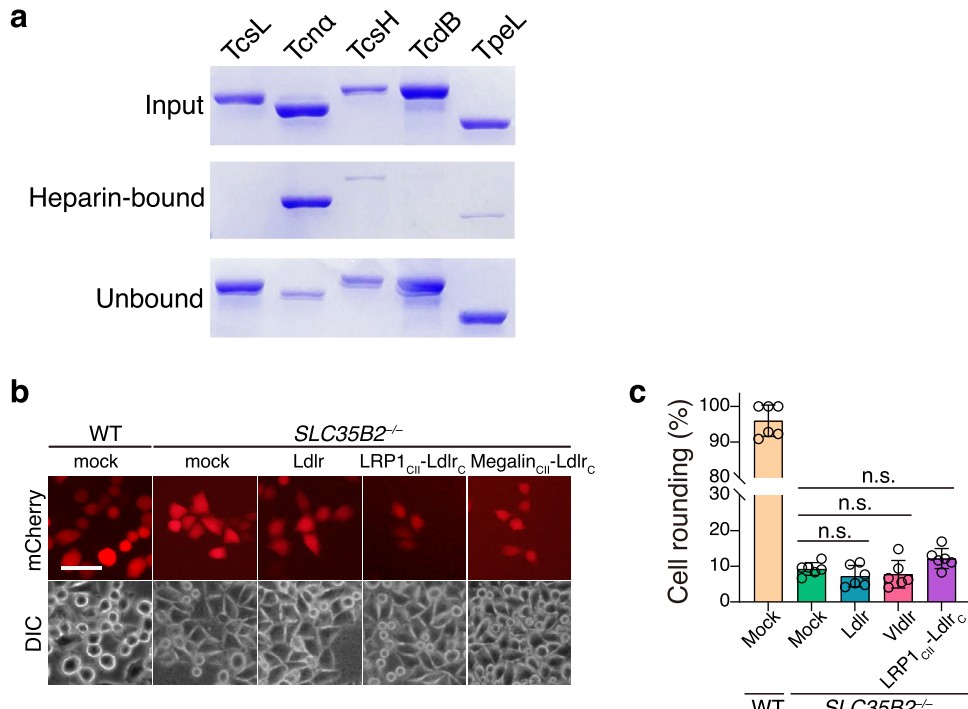

**Fig. 3 LRP1- and Megalin-mediated Tcnα uptake require cell-surface sGAG. a** TcsL, Tcnα, TcsH, TcdB, and TpeL were incubated with Heparin-Sepharose at 4 °C for 1 h. The protein samples were prepared from the input, toxin-bound heparin beads, and the supernatant was separated on an SDS-PAGE and detected by coomassie blue staining. **b** The HeLa WT or $SLC35B2^{-/-}$ cells were transfected with mock, Ldlr, LRP1$_{CII}$-Ldlr$_C$, and Megalin$_{CII}$-Ldlr$_C$. The cells were then incubated with 200 nM Tcnα on ice for 30 min, changed with the fresh medium, and incubated at 37 °C for 3 h. Representative images are shown. Red fluorescence (mCherry) marked transfected cells. The scale bar represents 50 μm. **c** The round-shaped cells among all mCherry-positive cells shown in **b** were quantified and plotted in a bar chart. Error bars ($n = 6$) indicate mean ± SD, n.s. not significant, two-sided Student's $t$-test.

that the LA domains of LDLR, LRP1, and Megalin are capable of recognizing Tcnα and mediating its entry. Owing to the similarity of these LA repeats, we propose that they may interact with Tcnα in a similar mode of action. However, other LDLR family members, such as Vldlr and ApoER2, failed to recognize Tcnα, indicating that the interactions between Tcnα and LDLR family proteins are somehow selective.

LDLR, LRP1, and Megalin can functionally mediate the cellular entry of Tcnα. According to the public datasets, both LDLR and LRP1 are widely distributed in various tissues including the liver and muscles, which are common targets for Tcnα. Megalin is expressed in limited organs like the brain and endocrine tissue, and its role in *C. novyi*-mediated pathology remains unclear. We notice that Megalin is highly expressed in Caco-2, a human colon carcinoma cell line that is widely used for studying LCTs. Thus, the role of Megalin in the cellular uptake of Tcnα, and potentially other LCTs, needs to be aware when Caco-2 cells are used. Since the expression levels of LDLR, LRP1, and Megalin varied in many different cells, such receptor redundancy may allow the toxin to target an extended range of host cells and bring advantages to the pathogen.

Direct interaction between LDLR/LRP1 and Tcnα seems to be weak, implying additional cellular factors/conditions may be involved. This is in line with our observations that sGAG-dependent toxin attachment is required for LDLR/LRP1-mediated entry of Tcnα. Synergistic actions between proteoglycans and LDLR family members are effective for the endocytosis of target ligands. While proteoglycans are normally abundant on the cell surface that can maximize the enrichment of ligands, LDLR family receptors can rapidly carry the cargoes into the endosomes[37–40]. This high-efficiency strategy is not only used for endogenous ligands such as remnant lipoproteins, amyloid-β, and PCSK[41–43] but also hijacked by various pathogens including

respiratory syncytial virus and TcdA[27,44]. Tcnα serves as another vivid example that a bacterial toxin uses such a "two-step" strategy to enter host cells. The identification of redundant entry receptors for Tcnα also increases our knowledge of LCTs. As the host receptors are demonstrated as keys to determining the pathology for LCTs[45–50], this study may further help to understand the pathogenesis of *C. novyi* infection-associated diseases.

## Methods

**Materials**. HeLa (H1, CRL-1958) and MCF-7 (HTB-22) cells were originally obtained from ATCC. MEFs (CTCC-003-0036), BJ (CTCC-400-0144), and U-87 MG (CTCC-ZHYC-0434) cells were purchased from Chinese Tissue Culture Collections (CTCC). Expi293F cells (A14527) were purchased from ThermoFisher Scientific. They were tested negative for mycoplasma contamination. HeLa cells were authenticated via STR profiling (Shanghai Biowing Biotechnology Co. LTD, Shanghai, China). Hela $LDLR^{-/-}$ and $SLC35B2^{-/-}$ cells were previously generated laboratory stocks[27,51]. All cell lines were cultured in DMEM media plus 10% fetal bovine serum (FBS) and 100 U penicillin/0.1 mg/mL streptomycin in a humidified atmosphere of 95% air and 5% $CO_2$ at 37 °C.

The following antibodies, reagents, and recombinant proteins were purchased from the indicated vendors: Alexa Fluor 488 goat anti-rabbit IgG (ab150077, 1:1000, Abcam), rabbit polyclonal IgG against β-Actin (ab227387, 1:5000, Abcam), rabbit monoclonal IgG against LDLR (ab52818 for western blot, 1:500; ab30532 for immunofluorescence, 1:200; Abcam), rabbit monoclonal IgG against LRP1 (ab92544, 1:20000 for western blot and 1:200 for immunofluorescence, Abcam), HRP-conjugated goat anti-human IgG-Fc antibody (SSA001, 1:3000, Sino Biological), Hoechst 33258 staining solution (E607301, BBI), NHS-Rhodamine fluorescent labeling kit (#46406, Thermo Fisher Scientific), recombinant human LRP1 Cluster II Fc chimera (R&D Systems, 2368-L2), Precast PAGE Gel (abs9309, Absin), Polyethylenimine Linear (PEI) MW25000 (40816ES03, YEASEN), and Heparin-Sepharose (Abcam, ab193268).

**Genes and cloning**. The DNA fragments encoding LRP1$_{CII}$, Megalin$_{CII}$, LRP1B$_{CII}$, and LRP1$_C$ were synthesized by a commercial vendor (Genscript, Nanjing). The DNA fragments encoding Ldlr, Ldlr$_{\Delta LA}$, Ldlr$_{\Delta LB}$, Vldlr, Lrp4, Lrp10, Lrp11, and ApoER2 were PCR amplified from Dharmacon™ cDNA/ORF Library and cloned

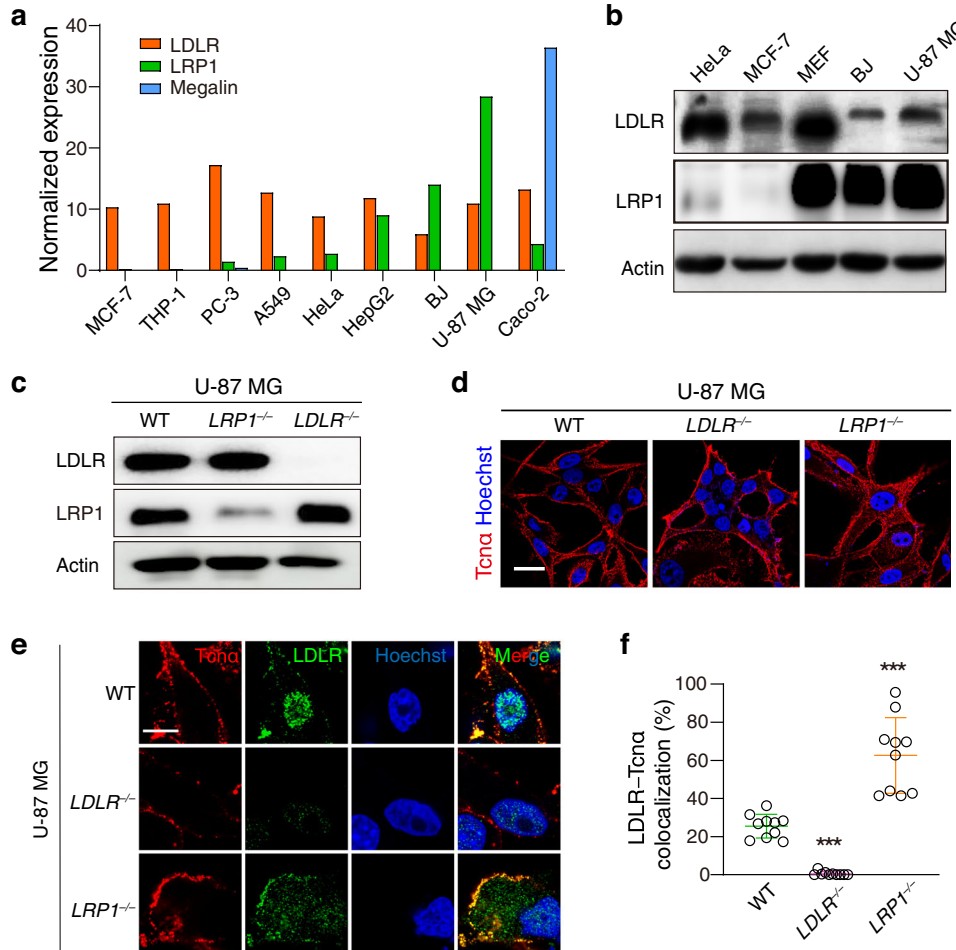

**Fig. 4 LDLR versus LRP1 in various cells. a** The mRNA levels of LDLR, LRP1, and Megalin in MCF-7, THP-1, PC-3, A549, HeLa, HepG2, BJ, U-87 MG, and Caco-2 cells are shown. Data were obtained from a public database (http://www.proteinatlas.org). **b** The protein levels of LDLR and LRP1 in the HeLa, MCF, MEF, BJ, and U-87 MG cells were measured by immunoblot analysis. **c** The depletion of LDLR and LRP1 in the U-87 MG $LDLR^{-/-}$ and $LRP1^{-/-}$ cells showed by immunoblot analysis. Actin served as a loading control. The experiments in **b**, **c** have been repeated independently twice with similar results. **d** Immunofluorescence analysis shows that Alexa Fluor 555-labeled Tcnα (50 nM) is robustly bound to the U-87 MG WT, $LDLR^{-/-}$, and $LRP1^{-/-}$ cells. Cell nuclei were stained with Hoechst dye. Representative images are shown. The scale bar represents 50 μm. **e** Immunofluorescent staining shows cellular localization of LDLR and endocytosed Tcnα in the U-87 MG WT, $LDLR^{-/-}$, and $LRP1^{-/-}$ cells. Cell nuclei were stained with Hoechst. Representative images are shown. Scale bars represent 10 μm. **f** Colocalization of LDLR and endocytosed Tcnα in the U-87 MG WT, $LDLR^{-/-}$, and $LRP1^{-/-}$ cells were analyzed by software ImageJ ver1.53. The percentage of the Tcnα signals that overlapped with LDLR in each cell was calculated and plotted as an open circle. Error bars ($n = 10$) indicate mean ± SD, ***$P < 0.001$ versus WT, two-sided Student's $t$-test.

into a pLVX-IRES-mCherry vector (Miaoling Bioscience & Technology Co., Ltd, P0424). The DNA fragments encoding $LRP1_{CII}$, $Megalin_{CII}$, and $LRP1B_{CII}$ were fused with the DNA fragment encoding $Ldlr_C$ or $LRP1_C$, followed by cloning into a pLVX-IRES-mCherry vector. The DNA fragment encoding $LDLR_{LA}$ was fused with the DNA fragment encoding human IgG Fc and cloned into a pHLsec vector for protein expression. $Ldlr_{\Delta NPxY}$ was generated by site-directed quick-change mutagenesis following the manufacturer's instructions (Agilent Technologies). All constructs were validated by DNA sequencing.

**Expression and purification of recombinant Proteins**. Recombinant Tcnα, TcdB, TpeL, TcsL, and TcsH were expressed in *Bacillus subtilis* SL401 and purified as His-tagged proteins[52]. In brief, *B. subtilis* cells were cultured at 37 °C till $OD_{600}$ reached 0.6 and then induced with 1 mM isopropyl-β-D-thiogalactoside at 25 °C for 20 h. The recombinant $LDLR_{LA}$-Fc with His-tag at C-terminus was expressed in the Expi293F cells. In brief, $5 \times 10^8$ Expi293F cells were transfected with 750 μg of pHLsec-$LDLR_{LA}$-Fc using 1 mg/ml PEI. The supernatant was collected 4 days post-transfection and applied to purification. All above recombinant proteins were purified by Ni-affinity chromatography and size-exclusion chromatography (GE Healthcare).

**Gene knockout in U-87 MG cell line**. To generate U-87 MG $LDLR^{-/-}$ cell line, the following sgRNA sequences were cloned into LentiGuide-Puro vectors (Addgene #52963) to target $LDLR$ genes: 5′-CCAGCTGGACCCCCACACGA-3′. To generate

U-87 MG $LRP1^{-/-}$ cell line, the following two sgRNA sequences were cloned into LentiGuide-puro-mKate2 vectors to achieve fragment knockout: 5′-CTGCCCAGA CGGATCTGACG-3′ and 5′-TGCGACTACGACAACGATTG-3′. Lentiviruses were generated by transfecting 293T cells with sgRNA plasmid, pSPAX2, and pMD2g. U-87 MG Cas9 cells were transduced with lentiviruses that express the sgRNAs. Mixed populations of infected cells were selected with puromycin (5 μg/ml). The KO efficiency of all mixed populations of KO cells was validated by immunoblot analysis.

**Cytopathic cell rounding assay**. HeLa and U-87 MG cells were transiently transfected using Polyjet following a manufacturer's instruction. Thirty-six hours post-transfection, the transfected cells were trypsin-digested and plated to the new 24-well plates. Cells were allowed to grow for additional 12 h and then applied to toxin treatment. The transfected HeLa $LDLR^{-/-}$ cells were exposed to a series of diluted Tcnα at 37 °C for 3 h. The transfected HeLa $SLC35B2^{-/-}$ cells were first incubated with 200 nM Tcnα on ice, changed with the fresh medium, and then incubated at 37 °C for 3 h. The U-87 MG cells were exposed to a series of diluted Tcnα at 37 °C for 20 h. The phase-contrast images of cells were then taken (Olympus IX73, 10× objectives). A zone of 200 × 200 μm was selected randomly, which contains 20–50 cells. Round-shaped and normal-shaped cells were counted manually. The percentage of round-shaped cells was analyzed using OriginPro (OriginLab, v8.5). All experiments were performed in three independent biological replicates. Statistical analysis was performed using OriginPro (OriginLab, v8.5).

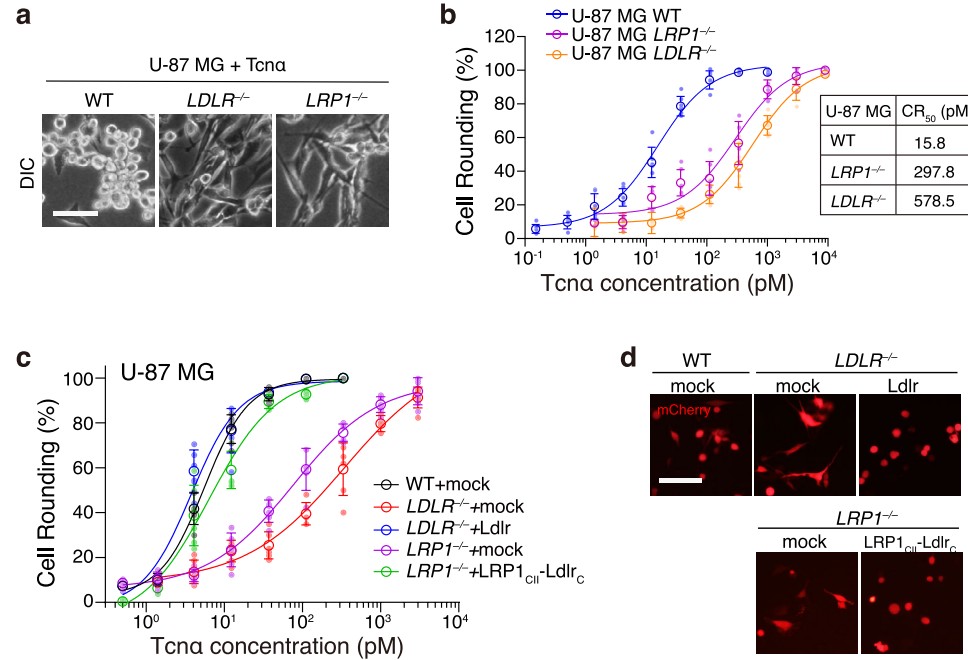

**Fig. 5 The U-87 MG LDLR−/− and LRP1−/− cells are more resistant to Tcnα. a** U-87 MG WT, *LDLR−/−*, and *LRP1−/−* cells were incubated with 30 pM Tcnα at 37 °C for 20 h. Cytopathic effect was observed in the U-87 MG WT cells but not the *LDLR−/−* and *LRP1−/−* cells using microscopic analysis for cell morphology. The scale bar represents 50 μm. **b** The sensitivities of the U-87 MG WT, *LDLR−/−*, and *LRP1−/−* cells to Tcnα were quantified using the cytopathic cell rounding assay. The percentage of round-shaped cells was measured and plotted on the chart. Error bars ($n = 6$) indicate mean ± SD. The $CR_{50}$ for Tcnα in the U-87 MG WT, *LDLR−/−*, and *LRP1−/−* cells were calculated and listed. **c** The indicated U-87 MG cells were transfected with mock, Ldlr, or $LRP1_{CII}$-$Ldlr_C$, followed by incubation with Tcnα for 20 h. The percentages of round-shaped cells are plotted on the chart. Error bars ($n = 6$) indicate mean ± SD. **d** The indicated U-87 MG cells were incubated with Tcnα (30 pM, 20 h) and the images were captured. Representative images are shown. Red fluorescence (mCherry) marked transfected cells. The scale bar represents 50 μm.

**Heparin-Sepharose pulldown assay**. Tcnα, TcsL, TcsH, TpeL, and TcdB were diluted into a final concentration of 0.5 μg/μl. Then 20 μl of toxin protein was incubated with 20 μl of Heparin-Sepharose (Abcam, ab193268) for 1 h a 4 °C. The Heparin-Sepharose beads were washed three times with PBS and collected as samples. All the samples were analyzed via SDS-PAGE analysis.

**Immunoblot analysis**. Cells were scraped and washed three times with PBS. Cell pellets were lysed with RIPA buffer (50 mM Tris, pH 7.5, 1% NP-40, 150 mM NaCl, 0.5% sodium deoxycholate, 1% SDS, protease inhibitor cocktail) on ice for 30 min. The protein amounts in cell lysate were determined by BCA assay (Beyotime, P0011). The cell lysates were heated for 5 min at 95 °C, analyzed by SDS-PAGE, and transferred onto a nitrocellulose membrane (GE Healthcare, 10600002). The membrane was blocked with TBS-T buffer (10 mM Tris, pH 7.4, 150 mM NaCl, 0.1 % Tween-20) containing 5% skim milk at room temperature for 1 h. The membrane was then incubated with the primary antibodies for 2 h, washed, and incubated with secondary antibodies for 1 h at room temperature. Signals were detected using the enhanced chemiluminescence method (Thermo Fisher Scientific, 34080) with GE imaging system AI680RGB.

**BLI assay**. BLI assay was performed with an Octet RED96e system and the data were analyzed with Octet Data Analysis software (Version:12.0.1.2, ForteBio, Fremont, CA, U.S.). In brief, 200 nM Fc-tagged proteins were immobilized onto capture biosensors (AHC biosensor, ForteBio) and balanced with binding buffer (20 mM Tris-HCl, 150 mM NaCl, pH = 7.4). The biosensors were then exposed to the indicated concentrations of Tcnα or RAP, followed by dissociation in the binding buffer.

**Dot-blot assay**. LRPAP1 and Tcnα of indicated amounts were spotted onto a nitrocellulose membrane and allowed to dry completely in the air. The membrane was blocked with 5% skim milk for 1 h at room temperature, followed by incubating with $LRP1_{CII}$-Fc/$LDLR_{LA}$-Fc/IgG Fc at room temperature for 4 h. The bound $LRP1_{CII}$-Fc/$LDLR_{LA}$-Fc was detected with a monoclonal antibody against human IgG Fc. For membrane EDC cross-link, after the $LRP1_{CII}$-Fc/$LDLR_{LA}$-Fc/IgG Fc incubation, the blots were further incubated with 5 mM EDC at room temperature for 1 h.

**Cell-surface toxin-binding assay**. Tcnα was labeled using an NHS-Rhodamine fluorescent labeling kit (#46406, Thermo Fisher Scientific) following the

manufacturer's instructions. U-87 MG WT, *LDLR−/−*, and *LRP1−/−* cells were incubated with 50 nM Rhodamine-labeled Tcnα in PBS for 30 min on ice. Cells were washed five times with ice-cold PBS and fixed with 4% paraformaldehyde (PFA) and the cell nuclei were labeled with Hoechst. Confocal images were captured with the Zeiss LSM 880 NLO with AiryScan System.

**Toxin internalization assay**. Tcnα were labeled using an NHS-Rhodamine fluorescent labeling kit (#46406, Thermo Fisher Scientific) following the manufacturer's instructions. HeLa or U-87 MG cells were incubated with 400 nM Rhodamine-labeled Tcnα in PBS for 30 min on ice, then washed three times with ice-cold PBS and incubated with the toxin-free medium at 37 °C for 10 min. Cells were washed with ice-cold PBS and subjected to immunofluorescence analysis.

**Immunofluorescence analysis**. Cells were fixed with 4% paraformaldehyde, permeabilized with 0.5% Triton X-100, blocked with 5% BSA, and then incubated with the LDLR antibody (ab30532, Abcam) overnight at 4 °C. The cells were then washed, incubated with the secondary antibody (goat anti-rabbit IgG Alexa488) for 1 h at room temperature, and stained with Hoechst for cell nuclei. Confocal images were captured with the Zeiss LSM 880 NLO with AiryScan System. Colocalization of Tcnα and LDLR was analyzed by the software ImageJ ver1.53.

**Statistics and reproducibility**. Data are presented as mean ± standard deviation (SD). The number of the sample size ($n$) and statistical hypothesis testing method (two-sided Student's *t*-test) are described in the legends of the corresponding figures. Statistical analyses of data were performed with GraphPad Prism v9.3 or OriginPro v8.5. *$P < 0.05$, **$P < 0.01$, ***$P < 0.001$, n.s. = not significant. For western blot analysis, the experiments have been repeated independently at least twice with similar results.

**Reporting summary**. Further information on research design is available in the Nature Research Reporting Summary linked to this article.

## Data availability
The source data behind the graphs and charts in the paper are provided as Supplementary Data. Uncropped blots are available in Supplementary Information.

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

## Acknowledgements

This study was partially supported by the National Natural Science Foundation of China (Grant no. 31970129 to L.T.). L.T. also acknowledges support from the Zhejiang Provincial Natural Science Foundation of China under Grant no. LR20C010001, Westlake Center for Genome Editing under Program no. 20200000A992210/001, Westlake Education Foundation, and Westlake Laboratory of Life Sciences and Biomedicine.

## Author contributions

Y.Z. and L.T. conceived the project and designed the experiments. Y.Z. and Danyang Li conducted most of the experiments. Y.Z., Dangyang Li, and A.C. purified the proteins and labeled the toxin. Diyin Li carried out the heparin-pulldown assay. L.H. and J.L. helped with the data analysis. Y.Z. and L.T. wrote the manuscript with input from all co-authors.

## Competing interests

The authors declare no competing interests.
