## [Peer Review File · Communications Biology]

Reviewers' comments:

Reviewer #1 (Remarks to the Author):

In this manuscript, Zhou et al., follow-up on their previous data, which suggested that LDLR family members other than LDLR may participate in the entry of Tcna. They show that LRP1 and Megalin serve as redundant receptors for Tcna and through deletion analysis, show that the LA repeats are involved in binding. Finally, they show distinct expression profiles of these receptors in a handful of immortalized cell lines and use this to suggest that Tcna may use different receptors to enter different tissues.

Overall, this manuscript is fairly well-written, and the experiments are all well designed. In general, this data falls short in supporting all of the conclusions of the paper. Additional experiments would help strengthen this work.

Specific comments:

- The majority of the cytopathic rounding (CR) data for the various constructs were presented at a single concentration (6nM) and either measured after 4h (Fig 1 and 3) or 3h (Fig 2). This is somewhat arbitrary and not consistent. Full dose titrations of each toxin on each cell type (at the same time point) is needed to give a sense of how each individual component contributes to the overall entry process. This is critical in my view.
- Generating truncations and deletions of receptors to identify which domain/region is involved in binding is an elegant way to show the role of a given domain, but requires controls to show that the generated constructs are functionally folded (as in Figure 1). Can the authors confirm that the LA domains are folded and/or bind Tcna through any other assays?
- Can the authors indicate through a figure perhaps what the sequence similarity is between each of the different LDLR LA domains is? This may help to explain why some work and other do not.
- The data in the manuscript shows that LDLR requires sGAG modification to facilitate toxin entry. Once again, it would be instructive and useful to know to what extent this is true quantitatively, by presenting full dose titrations.
- From figure 4e, it looks like LRP1 KO effects the expression or subcellular localization of LDLR, where LRP1 KO makes LDLR more membrane localized; similarly, from figure 4f, LDLR KO also seems to affect on the expression or subcellular localization of LRP1, where LDLR KO makes LRP1 express less. Can the authors comment on this? At face value, this would seem to indicate that there are a lot going on behind the LDLR family receptor biology, maybe it is cleaner to do a double KO?
- What is the Kd of LDLR, LRP1 and Megalin to Tcna? Or at least which one binds tighter to Tcna? This would be useful information to know.
- While showing the LRP1, LDLR and Megalin levels in different cell lines is interesting, these are all cell lines. When is the rationale for using these? It is not clear what relevance this has to disease mediated by C. novyi and especially which tissues in humans Tcna targets.

Minor comments:

- Reference 11 and 12 are the same reference
- Line 298, should read ..."three times...."

Reviewer #2 (Remarks to the Author):

Clostridium novyi α -toxin (TcnA) belongs to the family of large clostridial toxins, which modify Rho/Ras proteins by glucosylation or attachment of GlcNAc. Recently, Zhou et al. reported that TcnA acts synergistically through binding to glycosaminoglycans and to the LDL receptor. In the present paper, the authors extended their studies and report that the LRP1 and Megalin receptors are also able to participate in the uptake of TcnA.

This is an interesting paper extending our knowledge about TcnA. However, some statements are not clear and the results of the present study seem to be over-interpreted to some extent.

Specific comments

1. A major problem is the fact that for most studies a chimeric proteins of the LDL receptor and LRP1 and Megalin proteins were used. Therefore, it is not completely clear that LRP1 and Megalin are the receptors for TcnA. The data suggest this; however, it is not proven in a strict sense. Main parts of the chimeric receptors are from LDL. The kinetics of up-take and the synergistic effects might be different for complete LRP1 and Megalin receptors. These possibilities have to be discussed in detail and/or shown. It would be helpful if additional chimeric proteins are studied, antibodies are used or additional knock-out cells employed to strengthen the hypothesis that LRP1 or Megalin are receptors for TcnA.
2. In the title and the abstract, it should be mentioned that mostly not the full LRP1 or Megalin receptors were used but chimeric proteins were employed.
3. The binding data shown (e.g., Fig.4) are not supported by quantitative measurements and statistics.
4. Figure 5 is crucial for the conclusions of the whole paper. It is difficult to understand, why in LRP1-deficient cells, the dose-effect curve shift to the right is so strong although LDLR is present. Therefore, it would be supportive to show that the additional expression of LRP1 in the LRP1-KO cells shifts the dose effect curve back to the left.
5. I do not understand Fig. 4C. LDLR^{-/-} cells should not contain LDLR. The same problem is with the LRP^{-/-} cells. It appears that the indications are mixed up.
6. What was the TcnA concentration in Fig. 4D
7. Line 159: "MEFs was previous used..." please, correct.
8. Line 157: Please, correct the heading.

Response to Reviewers (COMMSBIO-21-3165)

We are grateful for the reviewers' time and suggestions. We are also sorry that the revision took a long time, partly due to some unexpected conditions during the pandemics. We have carried out additional experiments and added more discussion, which further strengthened our manuscript.

Reviewer #1

1) The majority of the cytopathic rounding (CR) data for the various constructs were presented at a single concentration (6nM) and either measured after 4h (Fig 1 and 3) or 3h (Fig 2). This is somewhat arbitrary and not consistent. Full dose titrations of each toxin on each cell type (at the same time point) is needed to give a sense of how each individual component contributes to the overall entry process. This is critical in my view.

Response: We thank the reviewer for pointing out the inconsistency of using different time points in the mentioned experiments, which may confuse the readers. In transiently transfected cells, the exogenous genes are expressed for a limited time. Therefore, we chose a short-time toxin-exposure method (3 hours) to assess the sensitivity of the transfected cells. By testing the sensitivity of HeLa WT and *LDLR*^{-/-} to Tcn α using this method, 6 nM was selected as the appropriate toxin concentration and was used in the following experiments. We have added the full dose titration curves at the same time point (3 hours post-Tcn α exposure) for these experiments. Also, we have updated the images (captured 3 hours post-toxin exposure) in Fig. 1 and the related Materials and Method section. The experiment in Fig. 3b was used to demonstrate the role of sGAG and a different toxin treatment method was applied; we have included the details in the text and corrected the legend for Fig. 3b. (Line 88-91, 312-318, Fig. 1b-c)

2) Generating truncations and deletions of receptors to identify which domain/region is involved in binding is an elegant way to show the role of a given domain, but requires controls to show that the generated constructs are functionally folded (as in Figure 1). Can the authors confirm that the LA domains are folded and/or bind Tena through any other assays?

Response: We thank the reviewer for the comment. We have performed immunoblot analysis to

demonstrate that the truncated/chimeric proteins, including Ldlr_{ALA}, Ldlr_{ALB}, LRP1_{CII}-Ldlr_C, Megalin_{CII}-Ldlr_C, LRP1B_{CII}-Ldlr_C, and Vldlr_{LA}-Ldlr_C, are well expressed in the transfected cells. (Supplementary Fig. 1a, b)

3) *Can the authors indicate through a figure perhaps what the sequence similarity is between each of the different LDLR LA domains is? This may help to explain why some work and other do not.*

Response: We thank the reviewer for the nice proposal. The basic structural unit of the LA domain is the LA module, and each LA domain contains multiple (the number is also different) LA modules. The binding specificity of a ligand is normally determined by one or two LA modules. However, we could not compare them at this stage because the specific toxin-binding LA module has not yet been defined.

4) *The data in the manuscript shows that LDLR requires sGAG modification to facilitate toxin entry. Once again, it would be instructive and useful to know to what extent this is true quantitatively, by presenting full dose titrations.*

Response: We thank the reviewer for the comment. This experiment was performed with a different method because the HeLa *SLC35B2*^{-/-} cells are much more resistant to Tcn α compared to the WT cells. In brief, these cells were pre-incubated with 200 nM Tcn α on ice for 30 minutes, changed with the fresh medium, and incubated at 37°C for 3 hours. Thus, this experiment was not feasible to be performed by toxin titrations, but we have added a quantitation as suggested by the reviewer. We have also included the details method in the text and corrected the legend for Fig. 3b. (Line 139-142, 316-318, Fig. 3c)

5) *From figure 4e, it looks like LRP1 KO effects the expression or subcellular localization of LDLR, where LRP1 KO makes LDLR more membrane localized; similarly, from figure 4f, LDLR KO also seems to affect on the expression or subcellular localization of LRP1, where LDLR KO makes LRP1 express less. Can the authors comment on this? At face value, this would seem to indicate that there are a lot going on behind the LDLR family receptor biology, maybe it is cleaner to do a double KO?*

Response: We agree with the reviewer that it could be an interesting observation. Based on our

immunoblot data, it seems that knocking-out LRP1 does not affect the total expression level of LDLR and vice versa. However, we cannot rule out the possibility that knocking out LRP1 will in some degree change the subcellular localization of LDLR. Like what the reviewer commented, there could be a lot going on behind the LDLR family protein biology. Besides, we have tried for several months to obtain the U-87 MG LDLR/LRP1 double KO cells but failed; probably the double KO of LDLR/LRP1 would cause severe defects in the U-87 MG cell's growth. (Line 169-170, Fig. 4c)

6) *What is the K_d of LDLR, LRP1 and Megalin to Tcna? Or at least which one binds tighter to Tcna? This would be useful information to know.*

Response: We thank the reviewer for the suggestion. The BLI assay can detect low-affinity interactions between Tcna and LDLR and between Tcna and LRP1. We tried to quantitatively measure the binding kinetics of LDLR-Tcna and LRP1-Tcna. For kinetic measurement methods such as BLI, SPR, or ITC, the ligand stock needs to be prepared at a concentration approximately 10-fold above the estimated K_d . However, a major problem is that all these proteins are technically difficult to be prepared at high concentrations, thus we were not able to obtain the K_d values for LDLR-Tcna and LRP1-Tcna. The current data suggest that the binding affinity between these molecules is low, likely other factors/conditions are required to facilitate the binding. (Line 132-134, Supplementary Fig. 2a, b)

7) *While showing the LRP1, LDLR and Megalin levels in different cell lines is interesting, these are all cell lines. What is the rationale for using these? It is not clear what relevance this has to disease mediated by C. novyi and especially which tissues in humans Tcna targets.*

Response: Both LDLR and LRP1 are widely distributed in various tissues including the liver and muscles, which are common targets for Tcna. Megalin is expressed in limited organs like the brain and endocrine tissue, thus its role in *C. novyi* mediates pathology seemed to be unclear. We have added this part to the Discussion section. (Line 224-227)

8) *Minor comments:*

Reference 11 and 12 are the same reference.

Response: We thank the reviewer for pointing this out. We have removed the repetitive reference.

9) Line 298, should read ..."three times...."

Response: As suggested, we have corrected the mistake. (Line 330)

Reviewer #2

Clostridium novyi α -toxin (TcnA) belongs to the family of large clostridial toxins, which modify Rho/Ras proteins by glucosylation or attachment of GlcNAc. Recently, Zhou et al. reported that TcnA acts synergistically through binding to glycosaminoglycans and to the LDL receptor. In the present paper, the authors extended their studies and report that the LRP1 and Megalin receptors are also able to participate in the uptake of TcnA.

This is an interesting paper extending our knowledge about TcnA. However, some statements are not clear and the results of the present study seem to be over-interpreted to some extent.

Response: We thank the reviewer for supporting our study. We have carefully revised the text to prevent the over-interpreted statements.

Specific comments

1) *A major problem is the fact that for most studies a chimeric proteins of the LDL receptor and LRP1 and Megalin proteins were used. Therefore, it is not completely clear that LRP1 and Megalin are the receptors for TcnA. The data suggest this; however, it is not proven in a strict sense. Main parts of the chimeric receptors are from LDL. The kinetics of up-take and the synergistic effects might be different for complete LRP1 and Megalin receptors. These possibilities have to be discussed in detail and/or shown. It would be helpful if additional chimeric proteins are studied, antibodies are used or additional knock-out cells employed to strengthen the hypothesis that LRP1 or Megalin are receptors for TcnA.*

Response: We agree with the reviewer that the reconstitutive chimeric proteins may not completely represent the biological properties of native proteins. On the other hand, LDLR family proteins are composed of several clear types of structural units and have high molecular weights (particularly for LRP1 and Megalin). Therefore, using chimeric proteins to study the roles of LDLR members in

the ligand-receptor interactions seems to be generally adopted. As an example, a recent study used chimeric LRP1 to investigate the cellular entry of the Rift Valley fever virus (Ganaie et al. *Cell* 2021). Following the reviewer's suggestion, we have added the essential discussion on it. We have also included LRP1_{CII}-LRP1_C as an additional control as suggested by the reviewer, which excludes the possible affection caused by the C-terminus of LDLR. Ectopic expression of LRP1_{CII}-LRP1_C also restored the entry of Tcna in the *LDLR*^{-/-} cells, which further strengthened our statement.

(Line 119-121, 203-207, Fig. 2a-c)

2) *In the title and the abstract, it should be mentioned that mostly not the full LRP1 or Megalin receptors were used but chimeric proteins were employed.*

Response: As suggested by the reviewer, we have indicated in the abstract that the reconstitutive proteins were employed in the study. (Line 23-25)

3) *The binding data shown (e.g., Fig.4) are not supported by quantitative measurements and statistics.*

Response: As suggested, we have quantified the measurements and added the statistics to this binding assay. (Fig. 4f)

4) *Figure 5 is crucial for the conclusions of the whole paper. It is difficult to understand, why in LRP1-deficient cells, the dose-effect curve shift to the right is so strong although LDLR is present. Therefore, it would be supportive to show that the additional expression of LRP1 in the LRP1-KO cells shifts the dose effect curve back to the left.*

Response: We thank the reviewer for the comment. A major reason for the strong curve shift for LRP1 KO in Figure 5 is that this experiment was performed in the U-87 MG cell line, which expresses medium to high levels of LRP1 and LDLR. This is why we picked this cell line to do the LDLR and LRP1 KO respectively. In the U-87 MG cells, both LRP1 and LDLR participate in the entry of Tcna. Besides, we have tried for several months to obtain the U-87 MG LDLR/LRP1 double KO cells but failed; probably the double KO of LDLR/LRP1 would cause severe defects in the U-87 MG cell's growth. As suggested by the reviewer, we further performed the rescue experiment and showed that LRP1_{CII}-Ldlr_C restores the susceptibility of the LRP1 KO cells to Tcna. (Line 188-

190, Fig. 5c, d)

5) *I do not understand Fig. 4C. LDLR^{-/-} cells should not contain LDLR. The same problem is with the LRP^{-/-} cells. It appears that the indications are mixed up.*

Response: We are sorry for the mislabeling. LDLR^{-/-} cells should be LRP1^{-/-} cells and *vice versa*. We have corrected the mistake. (Fig. 4c)

6) *What was the TcnA concentration in Fig. 4D*

Response: The concentration used in Fig. 4d is 50 nM, we have added the information to the manuscript. (Line 365, 615)

7) *Line 159: "MEFs was previous used..." please, correct.*

Line 157: Please, correct the heading.

Response: As pointed out by the reviewer, we have corrected these mistakes. (Line 160, 165)

Reviewers' comments:

Reviewer #1 (Remarks to the Author):

I wish to thank the authors for addressing many of my points. and for providing justification for when they did not directly address a particular point raised. The revised manuscript is greatly improved and is accepted as is.

Reviewer #2 (Remarks to the Author):

This is a revised version of a manuscript. The paper has clearly improved by the revision but the rebuttal is not totally satisfying. The authors explain that quantification is difficult, because, for example, the affinities of the receptors for the toxin appear to be rather low. Some more data about this point would have further improved the paper.

1. Line 47: This sentence might not be entirely correct. In addition, TpeL appears to modify Rho/Ras proteins by attachment of GlcNAc.
2. Line 143: "Together, these data suggest that cell surface sGAG facilitate LDLR-, LRPR-, and Megalin-mediated cellular entry of Tcnd." The data presented suggest that sGAG is essential for the action of the toxin and not only facilitate its action.
3. The impact of the various receptors appear to be different. For example, in U87 MG cells KO of LDL-receptor has a larger effect than KO of LRP although LRP is expressed at much higher concentrations.

Response to Reviewers (COMMSBIO-21-3165B)

We very much appreciate the reviewers for their time and support of this work. Their comments and suggestions greatly help us to improve the manuscript.

Reviewer #1

I wish to thank the authors for addressing many of my points. and for providing justification for when they did not directly address a particular point raised. The revised manuscript is greatly improved and is accepted as is.

Reviewer #2

This is a revised version of a manuscript. The paper has clearly improved by the revision but the rebuttal is not totally satisfying. The authors explain that quantification is difficult, because, for example, the affinities of the receptors for the toxin appear to be rather low. Some more data about this point would have further improved the paper.

Response: We very much thank the reviewer for the nice suggestion of further exploring the basis of protein interaction between Tcn α and LDLR/LRP1, yet these weak/unstable protein-protein interactions are challenging to be investigated (or potentially a cofactor is still unknown). Cross-linking is a powerful tool to investigate weak protein-protein interactions. Routine dot-blot assays showed no detectable signals for LRP1_{CII}-Tcn α binding. However, if the dot-blot assays were performed followed by EDC cross-link, obvious signals for LRP1_{CII}/LDLR_{LA}-Tcn α were detected. These data further suggest that the interactions between LRP1_{CII}/LDLR_{LA} and Tcn α could be weak/unstable. In addition, we are collaborating with the structural and biochemical labs, and hopefully will have a better view of how Tcn α interacts with LDLR/LRP1 in the future. (Line 135-140, Supplementary Fig. 2c-d)

1. *Line 47: This sentence might not be entirely correct. In addition, TpeL appears to modify Rho/Ras proteins by attachment of GlcNAc.*

Response: We thank the reviewer for pointing this out, and we are sorry for the inaccurate statement. TpeL can use either UDP-glucose or UDP-GlcNAc to modify Rho/Ras, while Tcn α only uses UDP-GlcNAc to modify Rho family members. We have modified the sentences and added the references for accuracy. (Line 47-49)

2. *Line 143: “Together, these data suggest that cell surface sGAG facilitate LDLR-, LRPR-, and Megalin-mediated cellular entry of Tcn α .” The data presented suggest that sGAG is essential for the action of the toxin and not only facilitate its action.*

Response: As suggested, we have changed the sentence to “Together, these data suggest that cell surface sGAG is essential for Ldlr-, LRP1-, and Megalin-mediated cellular entry of Tcn α .” (Line 148-150)

3. *The impact of the various receptors appear to be different. For example, in U87 MG cells KO of LDL-receptor has a larger effect than KO of LRP although LRP is expressed at much higher concentrations.*

Response: We thank the reviewer for the comment. We also notice that the U87 MG cells have a higher mRNA level of LRP1 than LDLR, while KO of LDLR has a larger effect than KO of LRP1 in these cells. However, because different antibodies were used for the detection of LDLR and LRP1, we are not able to directly compare the endogenous protein levels of LDLR and LRP1. Hence, here we remain conserved to make a statement.

REVIEWERS' COMMENTS:

Reviewer #2 (Remarks to the Author):

This is an important paper. The authors sufficiently addressed my comments. The paper has greatly improved.

Response to Reviewers (COMMSBIO-21-3165C)

We thank all the reviewers for their time and support of this work. Their advice and suggestions certainly help us to improve the manuscript.

Reviewer #2

This is an important paper. The authors sufficiently addressed my comments. The paper has greatly improved.